# The Complex Relationship between HTLV-1 and Nonsense-Mediated mRNA Decay (NMD)

**DOI:** 10.3390/pathogens9040287

**Published:** 2020-04-15

**Authors:** Léa Prochasson, Pierre Jalinot, Vincent Mocquet

**Affiliations:** Laboratory of Biology and Modelling of the Cell (LBMC), ENS de Lyon, Univ Lyon, CNRS UMR 5239, INSERM U1210, 46 allée d’Italie, 69364 Lyon, France; lea.prochasson@ens-lyon.fr (L.P.); pierre.jalinot@ens-lyon.fr (P.J.)

**Keywords:** HTLV-1, retrovirus, antiviral process, nonsense mRNA decay, UPF1

## Abstract

Before the establishment of an adaptive immune response, retroviruses can be targeted by several cellular host factors at different stages of the viral replication cycle. This intrinsic immunity relies on a large diversity of antiviral processes. In the case of HTLV-1 infection, these active innate host defense mechanisms are debated. Among these mechanisms, we focused on an RNA decay pathway called nonsense-mediated mRNA decay (NMD), which can target multiple viral RNAs, including HTLV-1 unspliced RNA, as has been recently demonstrated. NMD is a co-translational process that depends on the RNA helicase UPF1 and regulates the expression of multiple types of host mRNAs. RNA sensitivity to NMD depends on mRNA organization and the ribonucleoprotein (mRNP) composition. HTLV-1 has evolved several means to evade the NMD threat, leading to NMD inhibition. In the early steps of infection, NMD inhibition favours the production of HTLV-1 infectious particles, which may contribute to the survival of the fittest clones despite genome instability; however, its direct long-term impact remains to be investigated.

## 1. Introduction

HTLV-1 is a delta-retrovirus infecting approximately 10 million people worldwide [1]. Only 2%–5% of HTLV-1 carriers experience disease: either adult T cell leukemia/lymphoma (ATL), an aggressive form of leukemia characterized by the proliferation of CD4+ T cells, or HTLV-associated myelopathy/ tropical spastic paraparesis (HAM/TSP), an inflammatory disease associated with demyelination of the spinal cord [2,3,4]. HTLV-1 mainly infects CD4+ T cells, although it has the potential to infect a wide variety of other cells: CD8+ T cells, B lymphocytes, myeloid cells, endothelial cells, fibroblasts, neutrophils, monocytes, myeloid and plasmacytoids dendritic cells [5,6,7].

The roles played by innate and adaptive immune responses during HTLV-1 infection are extensively studied. Models of how HTLV-1 successfully escapes or dysregulates these immune responses in order to extend the infection and provoke diseases have been proposed and are regularly updated [8,9,10]. However, concomitantly with the initiation of the innate immune response and before the establishment of the adaptive response, HTLV-1 is targeted by several cellular host factors at different stages of the viral replication cycle. This intrinsic immunity is either constitutively expressed or may be closely coordinated with innate immunity. Although it is one of the first layers of defence from the host, our knowledge about its efficiency and its impact in the context of HTLV-1 is much less exhaustive. This intrinsic immunity relies on a large and diverse set of antiviral processes. Notably, several restriction factors such as the tripartite motif-containing protein (TRIM) and apolipoprotein B mRNA-editing enzyme catalytic polypeptide-like (APOBEC) family, SAM domain and HD domain-containing protein 1 (SAMHD1), miRNAs, BTS2/tetherin (recently reviewed here [11]) or zinc finger antiviral protein (ZAP) are already under investigation to evaluate their effect on HTLV-1 replication (Figure 1). First discovered and extensively studied in the case of HIV-1, there are fewer data available regarding HTLV-1 that are summarized in Table 1. However, the involvement of most of these factors remains controversial, and the exact role of the innate host defense against HTLV-1 is not clear, likely due to a lack of molecular studies on early infection stages.

The nonsense-mediated mRNA decay (NMD) pathway is an interferon (IFN) independent process that has been recently demonstrated by convergent studies to degrade several viral RNA [33,34,35,36,37,38,39,40,41], hence it can be considered as an antiviral process as well. Concerning HTLV-1, it was shown that NMD inhibition led to an upregulation in the steady state levels of all viral mRNAs, revealing a direct and/or indirect sensitivity to NMD [42]. A complementary study subsequently validated the direct involvement of NMD by measuring the stability of unspliced genomic RNA (gRNA), exposing an inverse correlation between the levels of NMD proteins and viral gRNA stability. The knockdown of NMD factors was also associated with an increase in the GAG protein encoded by the full-length gRNA [43]. Altogether, these data indicate that NMD targets HTLV-1 at translational level, thus confirming a function for NMD in host protection against pathogens (Figure 1). However, the persistence of an HTLV-1 infection around the world nowadays suggests that HLTV-1 succeeds in bypassing NMD, at least partially, similar to other antiviral processes.

In this review, we thus propose an analysis of the interplay between HTLV-1 and NMD. In order to give a better insight about this process since NMD does not have a wide audience in the HTLV-1 field to the contrary of other antiviral process mechanisms, we first describe the mechanism, its regulation and the RNA features triggering its initiation. Then, we analyse how HTLV-1 protects itself against NMD and suggest how the consequences of NMD inhibition might converge with known cellular dysfunctions associated with HTLV-1 infection, highlighting a relationship more complex than meets the eye.

## 2. NMD in the Cell

### 2.1. Mechanism and Regulation

NMD is a co-translational pathway conserved through evolution from yeast to mammals and leads to the degradation of mRNA. It is initiated when translation is terminated in a specific unfavourable environment. The slowing of the translation terminating steps due to mRNA sequence, organization and/or mRNP composition greatly favours the retention of the NMD central factor UPF1 at the stop codon with the stalled ribosome, leading to NMD activation. UPF1 is a helicase that binds RNA without sequence specificity. It appears to be displaced by the translating ribosome and subsequently accumulates on the mRNA 3′UTR [44,45]. It is not yet clear whether UPF1 plays a direct role in translation termination despite its association with the ribosome release factors eRF1 and eRF3, but it is an indispensable factor of NMD initiation. A recent study has shown that another NMD factor, UPF3B, can inhibit translation termination without UPF1 and favours post-termination complex dissociation in vitro [46]. UPF1 also interacts with UPF3B, mutually increasing the stability of them both at the stop codon. The interaction of UPF1 with the SMG1 PI3K kinase at the terminating ribosome completes the NMD initiating complex, which is named SURF for SMG1-UPF1-release factors [47,48,49]. Next, UPF1 phosphorylation is stimulated by the recruitment of UPF2 and the DEAH-box helicase DHX34 [50,51,52]. UPF2 may be recruited to the vicinity of UPF1 through an interaction with UPF3B [53,54]. UFP2 binds UPF1, inducing the large conformational change necessary for triggering UPF1 ATPase and helicase activities [55]. UPF1 hyperphosphorylation at its Nter and Cter (CH and SQ domains, respectively) creates scaffolds to recruit degradation-promoting factors such as the endonuclease SMG6 [56]. SMG6 cleaves RNA at the vicinity of the stop codon, leading to unprotected 5′ and 3′ RNA fragments, which are then degraded by the 5′–3′ exoribonuclease XRN1 and the exosome [57]. Phospho-UPF1 can also recruit the heterodimer SMG5-SMG7, which is directly linked to deadenylation (with CCR4-NOT) or de-capping (with DCP1a and DCP2) activity [58,59,60]. Well characterized in vitro, the in vivo steps necessary for UPF1 enzymatic activity have not yet been elucidated but could play a role in stripping proteins from the 3′ cleavage products to enable XRN1 exonuclease activity. UPF1 may also be involved in complex remodelling, controlling the necessary sequence of steps leading to decay [57]. In the last step, UPF1 is unphosphorylated by PP2A, serine/threonine-protein phosphatase 2A, and dissociates from the target mRNP [61].

Closely related to translation termination, NMD efficiency is naturally modulated by mRNP composition. Notably, the presence of an exon-junction complex (EJC) is a critical parameter: when located downstream of the stop codon, the EJC promotes the enrichment of the 3′UTR in NMD factors such as UPF3B and UPF2, leading to strong stimulation of NMD (reviewed in [62,63,64]). EJC was also described as an enhancer of SMG6-mediated endonucleolysis [65]. Moreover, different compositions of the EJC have been observed and are associated with NMD modulation [66]. While the EJC can be considered the most important NMD stimulator, multiple mRNAs can, nevertheless, undergo NMD in an EJC-independent manner. Currently, the sensitivity of these RNA is explained by the absence or the distance between NMD inhibitors binding motifs and the stop codon. For instance, converging studies demonstrated that the longer the distance between the stop codon and the poly (A) tail associated with the Poly(A) binding protein (PABPC) (referring to “long 3′UTR” RNA), the higher the rate of NMD initiation because PABPC stimulates translation termination and opposes UPF1 stable recruitment. Polypyrimide track binding protein1 (PTBP1), when associated with RNA downstream of a stop codon, inhibits NMD by preventing UPF1 recruitment [67]. It binds a specific sequence known as the RNA stability element, which was first identified in the Rous sarcoma virus RNA and more recently found in many cellular 3′UTR sequences close to the termination codon. Similarly, heterogeneous nuclear ribonucleoprotein L (hnRNPL) antagonizes NMD [68] (Figure 2).

According to the above description, any stop codon, as soon as it is read by the ribosome, might be able to initiate NMD, with the outcome being dependent on the equilibrium between the stimulators and the inhibitors at each step. While the unique function of NMD is to degrade mRNA, two main functions are usually associated with NMD depending on the origin of the stop codon initiating NMD: RNA quality control and gene expression regulation at the post-transcriptional level (for an exhaustive review, see [69]).

### 2.2. NMD Targets and Functions

Historically, NMD was described as a RNA quality control process, degrading mRNA with a premature termination codon (PTC). It is estimated that such RNA represents 5%–30% of human transcripts and may be the result of genomic mutations (nonsense and frameshift), faulty alternative splicing (approximately 30% of alternative splicing events) or translational errors. The truncated protein can be deleterious for the cell since it is non-functional, or it may even have a dominant negative effect. PTCs are present inside the open reading frame (ORF), creating a longer 3′UTR with a greater distance between the physiologic stop codon and the related translation termination regulatory environment. PTC may also be upstream of exon-exon junctions marked by an EJC. As described above, these parameters greatly favour NMD triggering and lead to the elimination of the PTC-harbouring mRNA.

In addition to the quality control function in which NMD prompts the cell to systematically degrade “aberrant” mRNA, NMD can also periodically regulate specific “non-aberrant” mRNA. This intermittent sensitivity to NMD is a programmed cellular protocol based on the local modification of an mRNA sequence, organization and/or mRNP composition. Among the different parameters influencing those local modifications, we can find uORF (upstream open reading frame), which is a translation that may trigger NMD due to the EJC downstream in the main ORF. CREB-2/ATF4 mRNA is a well-documented example [70,71,72]. Alternative splicing can also increase the 3′UTR length and modify the secondary structure of mRNA that are a major characteristic triggering NMD. It can also periodically produce alternative isoforms bearing a PTC. Several types of DNA repair or splicing factors, such as SR-proteins, are regulated like this, indicating an expanded impact of NMD on the post-transcriptional regulation [73,74,75,76]. Finally, programmed frameshifts (PRF) redirect the ribosome towards another reading frame with a PTC in more than 95% of the cases, as shown with the CC chemokine receptor (CCR) type 5 mRNA, as well as several interleukins [77]. In the following section we propose an analysis of the interplay between HTLV-1 and the NMD regarding these characteristics.

## 3. HTLV-1 Confronts NMD

### 3.1. HTLV-1 Viral mRNA Exhibits NMD-Initiating Features

HTLV-1 is a complex retrovirus with multiple proteins coded in a single genomic RNA. The overall sensitivity of HTLV-1 retroviral RNA to NMD has been known for a few years [42,43]. Although the molecular determinants of RNA features triggering NMD sensitivity have not yet been clearly defined [78], we can at least suggest that this sensitivity may have multiple origins; the viral RNA can be directly as well as indirectly targeted by the NMD machinery.

**Direct regulation**: As described above, the half-life of unspliced HTLV-1 RNA is increased under NMD inhibition. How can this sensitivity be explained? Since the mRNA is un-spliced, it should not have bound an EJC, especially in the 3′UTR, which is expected to hamper its sensitivity to NMD. However, according to the above findings and our knowledge of HTLV-1 RNA organization, the 3′UTR size stands out as a possible factor. While the median human 3′UTR size is ~750 nt [79], the gag mRNA 3′UTR is~4000 nt, making it a suitable target for NMD. For instance, Garcia et al. showed that two RNA(+) viruses had genomic and sub-genomic RNA NMD sensitivity because their 3′UTRs ranged from 1 kb to 2.5 kb [37]. In contrast, the secondary structure of RNA and its mRNP composition can also compete for the long 3′UTR to have an effect on sensitivity [40,67,80]. Another noteworthy parameter of HTLV unspliced mRNA, and that of other retroviruses, is the –1 frameshift (–1FS). HTLV-1 has two successive –1FS in the un-spliced mRNA region, allowing for the synthesis of three different polyproteins: GAG, GAG-PRO and GAG-PRO-Pol fusion proteins. The low frequency of these (–1FS) maintains the correct ratio of these three viral proteins. By slowing ribosome reading and preventing translation termination, a frameshift can promote NMD. On the other hand, Hogg et al. suggested that retroviral readthroughs and frameshiftings destabilize UPF1 accumulation in the 3′UTR and impair NMD by redirecting the ribosome and avoiding the in-frame stop codon, even at a rate of approximately 1% [81]. In the case of HTLV-1, the treatment of infected cells by okadaic acid led to increased levels of unspliced viral RNA associated with UPF1 [43]. Okadaic acid prevented the dephosphorylation of active UPF1 molecules and suspended NMD in the latest steps of mRNA decay. These observations strongly suggest that UPF1 is not stripped from the unspliced viral RNA and that this RNA is subjected to active decay.

**Indirect regulation**: In cells transfected with an HTLV-1 molecular clone, the knockdown of UPF1 led to a 3-to-4-fold homogeneous upregulation of all viral RNAs in the steady state [42]. This homogeneity might suggest that the transcriptional activator(s) of HTLV-1 can be controlled by NMD, leading to the indirect regulation of viral mRNA. Supporting this hypothesis, CREB-2/ATF4 has been shown to be involved in LTR transactivation, regulated by NMD and stabilized by Tax at the post-transcriptional level [42].

While the sensitivity of HTLV-1 to NMD is well documented, additional work is needed to clearly demonstrate which stop codons are the most likely to induce NMD. Moreover, to maintain its capacity to replicate, HTLV-1 had to evolve solutions to evade the NMD threat. These solutions are discussed in the next section.

### 3.2. HTLV-1 Protection against NMD

The arms race involving HTLV-1 and NMD led to the evolution of two viral countermeasures. The viral proteins Tax and Rex were shown to target the NMD process in what is called “trans-inhibition”, incapacitating the decay of viral as well as cellular NMD targets. This evidence emerged from observations that HTLV-1-infected lymphocytes were able to specifically stabilize globin mRNA with a PTC [42]. Nakano et al. also showed that, in HeLa cells co-cultivated with HTLV-1-infected cells, this inhibition was maintained as long as *Tax/Rex* mRNA was expressed [43].

Tax was first described as the main viral transactivator [82]. It has the capacity to bind multiple host factors in the nucleus and in the cytoplasm, leading to dysfunctions promoting cell transformation [83,84] Tax-dependent NMD inhibition was initially investigated due to its interaction with the translation initiation factor eiF3E/INT6 [85], known to interact with UPF2 and to be involved in NMD [86]. In addition to this interaction, lNT6 was observed to be delocalized from the nucleus to the cytoplasm by Tax. This study also revealed contacts between Tax and several NMD factors and a direct interaction between Tax and the helicase UPF1. A complementary study introduced interesting details on Tax: first, Tax can bind to the helicase domain of UPF1 at the exit of the RNA binding channel, preventing UPF1 loading onto its target. Second, when UPF1 is already bound to RNA due to its action in NMD, Tax binding blocks ATP hydrolysis and helicase activity, freezing UPF1 on RNA. These observations suggest a broad effect on UPF1 with the capacity to impact NMD at different steps [87]. When analysing viral mRNA, it is difficult to dissociate the transactivation role of Tax on the viral promoter from its post-transcriptional effect via NMD. Therefore, a mutant form of Tax specific for NMD interference must be engineered. Nevertheless, when Tax is expressed alone or from a provirus, the half-lives of host mRNAs, such as creb-2/atf4, growth and arrest DNA damage-inducible 45 (Gadd45A) and smg5, are stabilized as a consequence of NMD trans-inhibition.

The Rex protein was also shown to inhibit NMD. Similarly to Tax, several host mRNAs known to be NMD sensitive had increased half-lives upon Rex expression. Rex is known to bind viral RNA at the RxRE motif. Upon binding to RxRE, Rex controls viral mRNA splicing. It also contacts the CRM1 export system to ensure the nucleo-cytoplasmic shuttling of the unspliced viral mRNA [88,89,90]. To date, the mechanism of NMD inhibition by Rex has not been described. It has also not yet been investigated whether the HTLV-1 RNA secondary structure provides a first line of defence against NMD (Figure 3).

### 3.3. When Does NMD Inhibition Occur during HTLV-1 Infection?

During infection, HTLV-1 is spread in two different ways: viral propagation is initially dependent on cell-to-cell transmission, then it evolves towards polyclonal and monoclonal expansion (reviewed elsewhere [12]). Cell-to-cell infection depends on virion production. These virions are composed of structural proteins translated from singly spliced mRNA (ENV) and unspliced viral mRNA (GAG). Tax, as the viral transactivator, is indispensable for the production of this unspliced mRNA. Additionally, the modulation of splicing, leading to the stabilization of viral unspliced mRNA as well as their nuclear export, depends on Rex. By targeting the gag unspliced mRNA, NMD prevents virion formation. Supporting this hypothesis, knockdown of UPF2 was associated with increased levels of the p24 and p19 GAG protein [43]. Interestingly, HIV gag mRNA and GAG protein expression were also shown to be affected by UPF2 and SMG6 expression in the context of virus reactivation [91]. Hence, it is understandable that NMD, which does not require induction by type I interferon and thus acts as a cell-intrinsic antiviral barrier, plays a role in the early steps of HTLV-1 infection. The results of an analysis of viral mRNA kinetics showed a clear separation between the early phase when Tax and Rex (p21 and p27) are produced and the later phases characterized by the increase in other mRNAs, including gag mRNA [92]. This finding suggests that the stabilization of *gag* mRNA and the production of the gag protein necessitate the formation of a favourable environment: NMD inhibition might contribute to this initial condition. Notably, the results from kinetics experiments performed with RNA(+) virus infections suggest that the impact of NMD inhibition on viral RNA is greater in the early steps of the infection [37,39]. Finally, it is striking how the same factors, Tax and Rex, are involved in both the production and the protection of viral particles.

The second mode of provirus amplification is clonal expansion, which depends on the proliferation of a few selective clones. This oncogenic behaviour emerges from the modulation of multiple cellular processes by non-structural viral proteins such as Tax and HBZ; notably, Tax plays an essential role in this cellular transformation by generating instability and bypassing checkpoints: it inhibits DNA repair, disrupts cell cycle progression, and affects autophagy. It also modulates transcription through the modification of the epigenetic landscape and transcription complex composition and deregulates signalling pathways, including NF-κB (inducing its constitutive activation) and innate immune pathways (with an immunosuppressive effect) (reviewed in [4,12]). Knowing that Tax is the immunodominant HTLV-1 antigen in the T-cell response, it is considered that the progression of an infected clone is then favoured by the progressive inhibition or occasional bursts of expression of Tax [93], which confers a survival advantage through escape from the strong cytotoxic T-lymphocyte (CTL) immune response [94]. The other viral oncoprotein HBZ that triggers a less efficient immunity is continuously expressed in infected cells and plays important roles in viral latency and the proliferation of infected cells. Interestingly, *hbz* RNA appears to play specific roles in T cell proliferation (reviewed in [95]).

Although it is widely accepted that HTLV-1 infection leads to cell proliferation, it has been frequently observed to cause apoptosis and senescence of lymphoid and non-lymphoid cells in a Tax-dependent manner. It has been proposed that cells responding by senescence or apoptosis have very high levels of expressed Tax and Rex, are associated with robust replication of the virus and are subjected to great oncogenic pressure. This viral hyperactivity might be dependent on the site of integration, stimulating LTR transactivation and thus may be specific to each clone. Senescence induction has been shown to be linked with the stabilization of the p27 protein and cyclin-dependent kinase inhibitor p21CIP/WAF1 mRNA because of NF-κB hyperactivation ([96,97,98] reviewed in [99]). NF-B activation is a hallmark of Tax expression and HTLV-1-infected cells, and multiple steps in the NF-B pathway were shown to be regulated directly or indirectly by Tax. For example, NIK (MAP3K14), the NF-κB inducing kinase that activates IKKα, is highly expressed in HTLV-1-infected cells [100]. Moreover, p21, p27 and NIK transcripts are known to be NMD targets. Their mRNA levels are tightly controlled, and NMD inhibition experiments induced their stabilization [71,101,102]. In lung inflammatory myofibroblast tumors, decreased NMD leads to the increased expression of the transcript for the NIK protein kinase, which activates the NF-B pathway and promotes cytokine expression and inflammation. Supporting this observation, the stabilization of the NIK mRNA half-life was observed in a separate Tax and Rex-dependent manner [43,87]. Additionally, it is worth noting that NMD is involved in the apoptotic response induced by high stress levels; more precisely, it has been demonstrated that maintained NMD inhibition induces cell death [103,104]. Gadd45 isoforms promote apoptosis by upregulating the mitogen-activated protein kinase (MAPK) signaling pathway upon the activation of diverse forms of stress, including that from TNFα and DNA damage. Gadd45α and β transcripts are dependent on canonical as well as non-canonical NF-κB pathway activation and are upregulated in Tax and HTLV-1 expressing lymphocytes [105]. Gadd45α and β mRNA are also sensitive to NMD due to their respective 3′UTRs. This regulation of Gadd45 by NMD is evolutionarily conserved from flies to mammals. It was recently shown that the upregulation of Gadd45 isoforms due to NMD inhibition is a major contributor to NMD-associated programmed cell death [106]. Altogether, these observations suggest that, in the context of HTLV-1 infection, high levels of Tax and Rex, viral-induced senescence/apoptosis and NMD inhibition might be correlated (Figure 4). However, NMD inhibition in vivo does not always lead to the activation of apoptosis; for example, the absence of the key NMD factors UPF1 and SMG1 induces embryonic cell death, but UPF3-null mice are viable. Similarly, during hematopoiesis, NMD inhibition by UPF2 knockdown prevents hematopoietic stem cell and progenitor survival, whereas mature cells are only mildly affected [107,108,109,110]. Thus, whether NMD inhibition can activate apoptosis might be dependent on the degree of the NMD inhibition, the cell type and differentiation state.

While in the early steps of infection when HTLV-1 directly benefits from NMD inhibition to produce infectious particles, it is conceivable that, in a second step, this NMD trans-inhibition also participates in the selection of a clone with attenuated expression of Tax and Rex that is more suited to avoid the immune response and thus to maintain infection.

### 3.4. HTLV-1-Associated Pathologies. How Can NMD Inhibition Impact the Host in the Long Term?

As introduced above, HTLV-1 is the etiological agent of ATL and HAM/TSP. ATL is a malignant lymphoproliferative syndrome established after decades of latency and characterized by genetic instability combined with checkpoint adaptation. A favorable environment enabling malignant proliferation is dependent on the establishment of an immunosuppressive state. The viral protein Tax plays a major role in these steps. HAM/TSP is an immune-mediated inflammatory disease associated with the accumulation of HTLV-1-specific CD8+ T cells and infected CD4+ T cells in cerebrospinal fluid and neural tissues. It is characterized by a chronic inflammatory state due to elevated cytokine expression and production (reviewed in [111]). We wondered whether NMD downregulation, induced for the early infective stage of HTLV-1, may play a role in the later steps of the infection and could converge with ATL or HAM/TSP onset.

In 2015, genomic sequencing of 400 ATL samples showed that the mutation rate for ATL was relatively high compared to other hematologic malignancies, with an average of 2.3 mutations per mega base in coding regions [26,112]. Notably, transacting T cell-specific transcription factor GATA3, which is required for multiple steps of T-cell differentiation in both developing thymocytes and mature T cells, is commonly affected by nonsense and frameshift mutations. The authors suggest that these mutants confer altered protein function (possibly dominant negative functions due to truncated proteins), rather than GATA3 haploinsufficiency. Most CCR4 and CCR7 mutations in ATL-related proteins cause truncation of the cytoplasmic domain with gain of function. Moreover, more than half of ATL cases have either nonsense or frameshift mutations in the components of the class I MHC. Although the NMD sensitivity of mRNA resulting from these hotspot mutations has not yet been analysed, these mutations seem to be promising targets, and NMD inhibition by Tax and Rex might be an important parameter to consider. Of course, due to its quality control function, NMD defects contribute to genetic instability; in combination with alternative splicing, NMD tunes the level of many DNA repair factors [75,113,114]. Pancreatic squamous carcinoma cells have mutations in the *upf1* gene allowing the synthesis of mutated dominant negative p53, functionally correlating NMD inhibition with cancer [115]. Loss-of-function or overexpression of NMD proteins is also associated with several other cancer types, including colorectal cancer, hepatocellular carcinoma and neuroblastoma [116,117,118]. NMD is also involved in the adaptation to stress response [119,120]. Hypoxia, amino acid depravation and reactive oxygen species production downregulate NMD, which leads to the stabilization of transcripts such as activating transcription factor ATF4, ATF3, ATF6, CCAAT-enhancer binding protein homologous protein (CHOP) and TNF receptor-associated factor 2 (TRAF2), which re-establish homeostasis by the integrated stress response (ISR). It is now acknowledged that tumour cells must adapt to microenvironment stresses such as these to proliferate. In this context, it has been shown that NMD downregulation plays a role in this adaptation, promoting tumorigenesis [121]. Moreover, it was also recently proposed that NMD tunes the immune response. Impaired NMD in mice with forebrain-specific UPF2KO triggers immune response activation and results in exacerbated neuroinflammation. The latter symptom was partially reversed upon UPF2 restoration [122]. In Arabidopsis, NMD downregulation due to bacterial infection has been shown to control the turnover frequency of numerous TIR domain-containing, nucleotide-binding, leucine-rich repeat (TNL) receptor mRNAs, inducing innate immunity. However, maintained NMD inhibition by silencing NMD components deregulates homeostasis, leading to an autoimmunity phenotype characterized by stunting, spontaneous formation of necrotic lesions, and elevated salicylic acid levels [123]. The role of NMD in cancer and immunity likely depends on the tissues implicated and the associated genomic stress, but the effects of its inhibition in HTLV-1 unrelated examples likely converge with HLTV-1 phenotypes.

However, to clarify whether there is an effective link between NMD and HTLV-1-associated pathologies, the status of NMD in the later stages of the infection and in patient cells, ATL and HAM/TSP, has yet to be analysed. The absence of (+) strand transcription in most ATL cells due to epigenetic repression and genomic alterations raises the following questions: Is the burst of (+) strand expression sufficient to inhibit NMD? HBZ, the protein expressed from the HTLV-1 minus strand, is a critical component of cell proliferation and tumorigenesis and maintains constant expression during infection, in contrast to Tax and Rex; could it also be involved in NMD inhibition? NMD is also sensitive to bivalent cation concentrations [124], and interestingly, ATL patients show hypercalcemia; does this suggest that NMD can be constitutively inhibited? Without additional experimental data, these questions remain unanswered.

## 4. Concluding Remarks

Although HTLV-1 infects its host on the long term and can spread into the organism by inducing clonal expansion of infected cells, a strong immune response is usually directed against some of its protein. Over the years, models were proposed to explain this apparent contradiction. However, in these models, the intrinsic immunity operating within virus-infected cells, i.e., antiviral factors, are usually put aside. The very persistence of the HTLV-1 infections suggests that these mechanisms fail to restrict the virus or that HTLV-1 evolved countermeasures, if not completely, at least efficiently enough to allow minimum expression. In this review we have presented that HTLV-1 depends on a polycistronic RNA, which tends to have a long 3′UTR which is a critical NMD determinant. To protect its mRNA, HTLV-1 evolved independent mechanisms to trans-inhibit NMD through two viral proteins: Tax and Rex. The strength of these countermeasures shows the size of the threat that NMD constitutes for viral replication and the place of NMD among the cellular antiviral function. Those conclusions are also supported by comparison with several other viruses, especially retroviruses and (+) RNA viruses, which RNAs are also targeted by NMD and that evolved protective means [33,35,36,37,39,40,41,42,43,87,125,126,127]. It is important to note that the viruses that develop a trans-inhibition process against NMD, such as HTLV-1 and potentially HIV, might only protect themselves progressively; before the viral inhibitor is expressed at an adequate level and while viral mRNA are at low level during the early phase of infection, the NMD would inhibit optimal viral replication as demonstrated with the mouse hepatitis virus (MHV, a prototypic member of the betacoronaviridae, as SARS-CoV and MERSCoV), the plant virus potato virus X (alfaflexiviridae) and turnip crinkle virus (Tombusviridae) [37,39]. The importance of this initial slowdown of viral replication for the survival of the host on the long term is still unknown but deserves further investigations.

However, while HTLV-1 and the host cell seem to be engaged in an arms race involving NMD, we wonder which partner truly benefits from NMD inhibition during each steps of the infection. The virus, in the early steps, because it is able to produce viral particles again? The host, since NMD inhibition establishes the immune response? Is the apoptosis induced from NMD inhibition contributing to the clonal selection involved in leukaemia emergence? Could the NMD activity be controlled by an occasional burst of Tax expression and be involved in the latency exit as suggested with HIV [91]? Similarly, does NMD inhibition contribute to the genetic instability and cell adaptation associated with HTLV-1, as observed in NMD-related cancers? All these questions have yet to be addressed in detail. It is also rather interesting to note that these questions could be asked in the context of other antiviral processes where their activation could help HTLV-1 during clonal expansion. For instance, it was observed that APOBEC family members generate nonsense mutations on the (+) strand of the viral genome which favour the escape of HTLV-1-infected cells from the host immune system, while the HBZ (-) strand is not affected. Supporting that point, APOBEC family members seem to be more involved in the ATL progression than HAM/TSP in the actual literature. 

Finally, multiple “NMD-like” processes have been recently discovered, and they depend on the RNA helicase UPF1 and factors recognizing specific RNA features. In addition to NMD, UPF1 is involved in replication-dependent histone mRNA decay, miRNA decay, glucocorticoid receptor-mediated mRNA decay and regnase1-mediated mRNA decay (RMD) (reviewed in [128]). Since evidence has shown that Tax inhibits UPF1 enzymatic activity by preventing its association with RNA [87], we can hypothesize that other RNA decay pathways that depend on UPF1 are also likely altered, extending the impact of UPF1 on HTLV-1 deregulation. Notably, RMD plays a critical role in immunity and inflammation with the downregulation of multiple transcripts, such as IL-6, IL-2, IL-1b, tumour necrosis factor receptor 2 (TNFR2), CD44, and the proto-oncogene c-Rel, which are also often characterized in HTLV-1 infection [111,129].

Considering HTLV-1 has already evolved counteracting measures to take advantage of the host machinery, it is still a promising strategy to gain insights into these molecular biology processes to better understand the virus biology and eventually develop efficient antiretroviral therapies.

## Figures and Tables

**Figure 1 pathogens-09-00287-f001:**
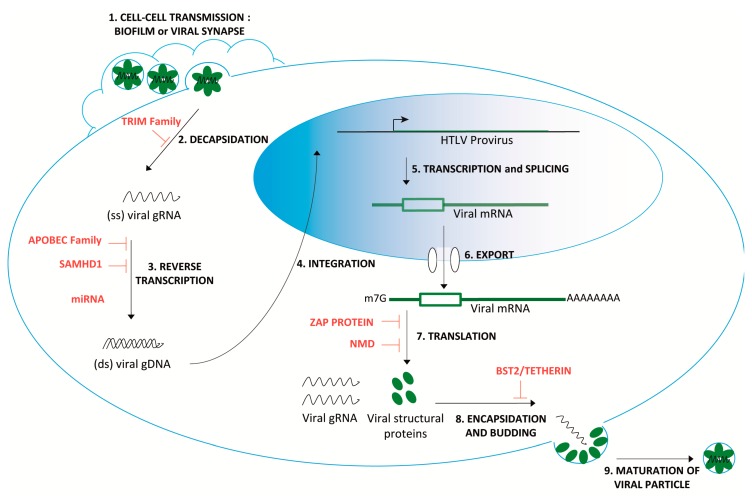
HTLV-1 infection cycle and host antiviral factors. After attachment to the cell membrane receptors (GLUT1, HSPG or NRP-1) through the viral Env protein, a fusion process enables the release of the capsid core containing the viral genome and the proteins into the cytoplasm [12]. Immediately after cell entry and before reverse transcription, the tripartite motif-containing protein (TRIM) family antiviral factors could limit the decapsidation step by recognizing determinants of the capsid, although it has not yet been investigated at the mechanistic level for HTLV-1 (Table 1). Following the entry, the viral RNA genome is reverse transcribed into double-stranded DNA, and together with associated proteins, this newly synthesized DNA forms the reverse transcription complex, also called the pre-integration complex. This reverse transcription step could be restricted early by SAM domain and HD domain-containing protein 1 (SAMHD1), decreasing the available pool of cellular dNTPs, by apolipoprotein B mRNA-editing enzyme catalytic polypeptide-like (APOBEC) members that can misedit the HTLV-1 RNA genome and as recently suggested, by miRNA that likely prevents the formation of the pre-integration complex (Table 1). To date, no antiviral factors have been fully characterized regarding the integration, transcription and splicing or RNA export steps. However, viral mRNA can be targeted at the translational level by the zinc finger antiviral protein (ZAP) protein, as well as by the nonsense mediated mRNA decay (NMD) process (described in the following sections). Finally, bone marrow stromal cell antigen 2 (BST2) could tether nascent virions at the budding step before the release and maturation of the viral particle.

**Figure 2 pathogens-09-00287-f002:**
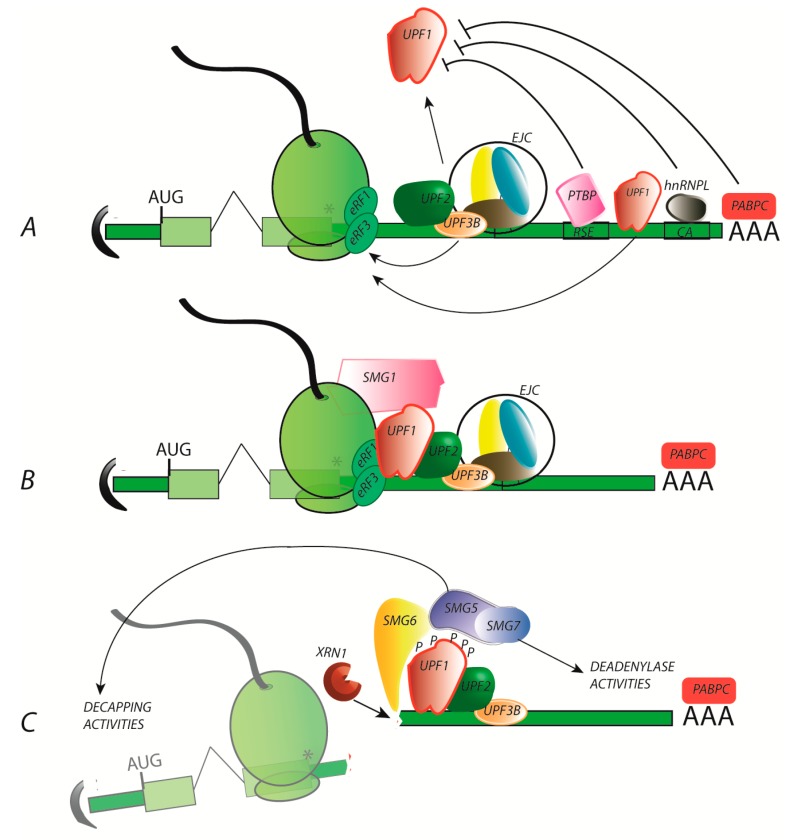
The nonsense-mediated mRNA decay (NMD) mechanism in three key steps. The NMD mechanism is organized into approximately 3 key steps. (**A**) Translation termination delay: NMD is initiated when translation termination is delayed. Multiple factors can be involved, notably by preventing/promoting the recruitment of the RNA helicase UPF1 to the ribosome. (**B**) UPF1 recruitment and stabilization favours the assembly of an active NMD complex. The EJC is a very potent stimulator of NMD initiation, bringing UPF2 and UPF3 in close vicinity to UPF1. Although EJC-independent NMD events occur, the details of the mechanism continue to be debated. (**C**) UPF1 phosphorylation by SMG1 initiates the decay step with the recruitment of the endonuclease SMG6 or/and the SMG5/SMG7 complex. SMG6 cleavage is followed by XRN1 degradation, while SMG5 and SMG7 initiate de-capping and deadenylating activities, respectively.

**Figure 3 pathogens-09-00287-f003:**
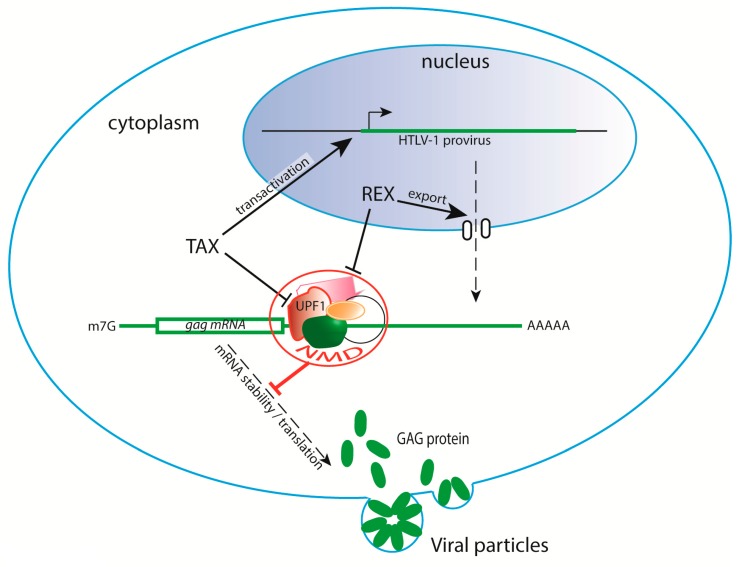
HTLV-1 confronts NMD. NMD is able to target viral gag mRNA, preventing further formation of viral particles. However, the viral proteins Tax and Rex, which are involved in viral transactivation and unspliced viral mRNA nuclear export, respectively, have been shown to inhibit NMD. The Rex mechanism of action has yet to be deciphered, while several approaches have revealed that Tax was shown to target UPF1.

**Figure 4 pathogens-09-00287-f004:**
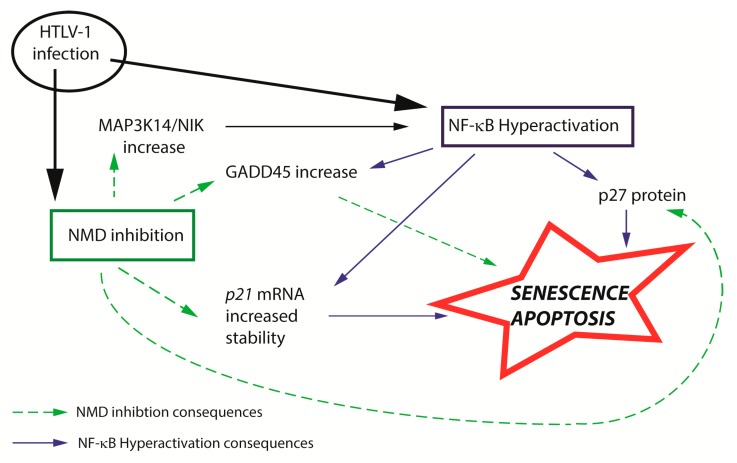
Convergence of NMD inhibition and NF-κB hyperactivation. HTLV-1-induced NMD inhibition and NF-κB hyperactivation might induce apoptosis and senescence via the stabilization/stimulation of the same host factors.

**Table 1 pathogens-09-00287-t001:** Potential involvement of host antiviral factors in HTLV-1 restriction. Regarding TRIM Family implication, all data reinforce the potential involvement of the TRIM family in the early restriction of HTLV-1 replication. The majority of the data are obtained by whole exome sequencing analysis [13] or microarray analysis of CD4+ T Cells [14] in HAM/TSP cohorts of patients. Consistently with the fact that HTLV-1 mainly replicates by clonal expansion and the fact that virions are poorly infective, the involvement of APOBEC family, as well as BST2/Tetherin, remains uncertain. HTLV-1 seems to be more resistant to h3AG than HIV-1, but interestingly h3AG could generated nonsense mutations in the viral genome that enable HTLV-1-infected cells to escape from the host immune system in ATL cases. SAMHD1 is able to restrict a broad range of retroviruses [15], including HIV-1, but further studies are needed to assess its implication in HTLV-1 restriction and/or pathogenesis. As previously demonstrated for HIV-1 [16], the role of miRNA in HTLV-1 infection has recently emerged [17]. Finally, HTLV-1 is susceptible to a ZAP-mediated regulation, which was shown as an mRNA decay pathway depending on the translation like the NMD pathway [18]. These data regarding HTLV-1 restriction by the host antiviral factors are quite still conflicting. It could be explained in part by the fact that a lot of these studies are based on data obtained in HIV, where the molecular determinants of the retroviral replication and life cycle differ from HTLV-1. Secondly, it could be explained by the fact that all these process are differentially involved according to the cell-type infected, the stage of infection, or even the evolution of chronically infected patients towards HAM/TSP or ATL.

Host Antiviral Factors	Restriction Mechanisms	Potential Involvement in HTLV-1 Restriction		References
**TRIM Family**	Recognition of the capsid lattice Interference with disassembly of the viral particles. E3 ubiquitin ligase activity targeting viral core component. [19]	TRIM19/PML interferes with HTLV-1 replication via the proteosomal degradation of the viral protein Tax.	+	[20]
		R136Q TRIM5α variants are correlated with lower proviral loads.	+	[13]
		Negative correlation between HTLV-1 virological parameters or clinical status and the expression of TRIM5α and TRIM22. Negative correlation between Tax mRNA levels and the expression of TRIM19/PML.	+	[14]
**APOBEC Family**	C->U editing enzymes leads to stopping codon incorporation after reverse transcription.	APOBEC3G (h3AG) is packaged into HTLV-1 virions. The HTLV-1 genome may be edited in vivo by h3AG, as well as other hA3 members (A, B, C, F, and H).	+	[21,22,23]
		G-to-A mutations were not detected in the proviruses from infected patients. HTLV-1 Gag protein would reduce the packaging of h3AG into virions.	-	[21,24]
		Loss-of-expression mutations likely generated by hA3G induced mutagenesis in HTLV-1 genes of 60 ATL cases.	+	[25]
		hA3B increased expression in ATL and asymptomatic carriers.	+	[26]
		No correlation with the clinical HAM/TSP states, PVLs or viral mutations.	-	[13]
		Negative correlation between members of APOBEC3 family and Tax mRNA levels.	+	[14]
		In HTLV-1 infected humanized mice showing ATL-like feature, an increased expression level of hA3. No correlation in HAM/TSP cohorts study.	+/-	[27]
**SAMHD1**	SAMHD1 decreases the pool of available dNTPs, inhibiting reverse transcription. Modulates antiviral activity by inhibiting the NF-κB and interferon pathways [28]	No effect on HTLV-1 and Tax expression in macrophages and cycling CD4+ T cells.	-	[15]
		SAMHD1-mediated apoptotic response in human primary monocytes.	+	[29]
		No correlation with the HAM/TSP parameters.	-	[13]
		SAMHD1 is negatively correlated within HAM/TSP cohorts.	+	[14]
**miRNA**	miR-28-3p targets genomic viral mRNA of HTLV-1 strains.	miR-28-3p could inhibit HTLV-1 replication by reducing expression of all viral proteins in a Tax-independent manner.	+	[30]
**BST2/Tetherin**	Inhibits the release of enveloped viruses by tethering them to the cell surface, where they undergo endocytosis and degradation.	No correlation with cell-cell transmission.	-	[31]
		No correlation with the HAM/TSP parameters.	-	[13]
		BST2 is negatively correlated with HAM/TSP status.	+	[14]
**Zinc finger Antiviral Protein (ZAP)**	Targets viral RNA at specific response elements (such as ZRE). Recruits cellular decay factors (PARN, DCP-1, 3‘–5’ exosome). [18]	HTLV-1 is susceptible to a ZAP-mediated viral RNA processing during early infection.	+	[32]

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
