# Peer review of "The Complex Relationship between HTLV-1 and Nonsense-Mediated mRNA Decay (NMD)"

_pathogens, 2020, doi:10.3390/pathogens9040287_

Round 1

Reviewer 1 Report

Lea et al. review the role of nonsense-mediated decay (NMD) in the context of HTLV-1 infection.

The review is divided into a short introduction on intrinsic/innate antiretroviral immunity, a section summarizing our current understanding of NMD, and a final section on HTLV and NMD.

The first part of the review is not particularly useful, there are some citation issues, and the net relevance of this section is unclear considering the authors’ speculation that NMD may or may not be antiviral in the context of HTLV. However, the second two sections are interesting and well-written. NMD is described in the context of viral infection and data supporting a role for NMD in regulating HTLV replication is discussed. One critique though is that the review could do a better job of identifying/proposing key new research questions; this aspect should be improved.

Major comments as follows:

1. The general premise that NMD is an antiviral process in the context of HTLV or HIV is controversial, and the authors are, appropriately, open to the possibility that it is not. Accordingly, a better title might cut the antiviral bit and just be “The complex relationship between HTLV-1 and nonsense-mediated mRNA decay”. The first sections that coach NMD in the context of other antiviral processes could be synthesized or removed, don’t really serve a useful purpose especially in the context of other recent reviews on restriction factors.

2. References for the first section are suboptimal; 2-4, 6, 12, 14 may not be the best; Forlani et al. repeated. Should be carefully curated. Key papers (57 and 58) are hinted at early (abstract and intro) and should be cited / named here.

Minor comments:

1. Note- in most instances TRIM5a’s ability to block HIV is species-specific (p. 3).
2. APOBEC3 causes G-to-A mutations through conversion of cytosine to uracil (not the other way around; p. 4)
3. Should better explain Tax-driven clonal expansion (p. 5)
4. How does lack of BST2 polymorphs in HAM/TSP eliminate it as a restriction factor for HTLV (p.6)
5. Fig. 1- does NMD block translation? More analogous to ZAP?

Reviewer 2 Report

The review article “An antiviral process targeting HTLV-1: the complex relationship between HTLV-1 and nonsense-mediated mRNA decay” by Lea et al., discussed the various host machinery pathways to degrade viral RNA and elaborating on NMD (nonsense-mediated RNA decay mechanism) for HTLV-1 infection. The review is very helpful as a collection of the very few articles available on the subject along with a comparison to the HIV infection mechanism. The review needs a major revision of the language and seems to be written as a collection of information without any concrete organization.

  1. The authors should consider serious revision of the manuscript for English grammar as there are many complex sentences that are hard to understand for the readers.
  2. The authors should rectify the reference formats as they have used two different forms of the bracket for the references.
  3. The authors should not label the figure number with the heading but add it to the text at relevant places.
  4. The review lacks analysis and collective interpretation of the published findings but literally states the findings from each article in an active voice. The authors should consider collecting the findings from the studies and summarizing them.
  5. The review lacks and future perspectives or conclusions from the author's viewpoint on the field which is lacking in the field currently.
  6. The authors should attempt to organize the information and reduce the length of the manuscript to make it focused on the topic.

Reviewer 3 Report

Comments to the Author

The authors’ objective seems to be to summarize the antiviral mechanisms for HTLV-1 infection in infected host cells using the available literature. The authors first describe the host restriction factors involved in the inhibition of the anti-HTLV-1 process. Thereafter, the NMD-related molecules and their functions are presented. The last part describes NMD inhibition by HTLV-1-related molecules. However, as the manuscript contains many descriptions that do not support the authors’ objective, the molecules and mechanisms that are important for the anti-HTLV-1 process are not evident. Additionally, figures 2 and 3 are not used to reinforce the text. Since most abbreviations are used without any definition, the reader has to search those abbreviations on all such occasions. Collectively, this manuscript does not achieve its objective.

Round 2

Reviewer 2 Report

The authors have addressed my prior concerns.

Author Response

9/04/2020

Revision answers;                   

Dear reviewer, we are very grateful for these last comments concerning minor corrections.

We modified the text as suggested:

-Page 4 line 11. “..., suggest that HLTV-1 succeeds...” has been changed for ‘..., suggests that HTLV-1 succeeds...’

-Page 2 line 48, Page 4 line 9, Page 5 line 52, and Page 6 line 95, space prior to citation has been added.

-Page 7 lines 160. The expanded name of GADD45 was used at its first occurrence

Kind regards

Vincent Mocquet

Reviewer 3 Report

This manuscript has been improved, in particular by modification of organization. The description of the interplay between NMD and HTLV-1-related molecules has been polished in the revised version. There are a few minor errors which need to be corrected:

Page 4 line 11. “..., suggest that HLTV-1 succeeds...” should be ‘..., suggest that HTLV-1 succeeds...’.

Page 2 line 48, Page 4 line 9, Page 5 line 52, and Page 6 line 95. Space prior to citation should be added.

Page 7 lines 160. Abbreviation of GADD45 was used prior to expanded name (Page 9 lines 225-226).

Author Response

(The authors gave the same response as above.)
